# Color Tuning by Oxide Addition in PEDOT:PSS-Based Electrochromic Devices

**DOI:** 10.3390/polym11010179

**Published:** 2019-01-21

**Authors:** Delphin Levasseur, Issam Mjejri, Thomas Rolland, Aline Rougier

**Affiliations:** 1CNRS, Univ. Bordeaux., ICMCB, UMR 5026, Bx INP, F-33600 Pessac, France; d.levasseur@ast-innovations.com (D.L.); mjejri08@gmail.com (I.M.); thomas.rolland@icmcb.cnrs.fr (T.R.); 2Aquitaine Sciences Transfert, 33405 Talence, France

**Keywords:** Electrochromic, display, PEDOT:PSS, Fe_2_O_3_, color-tuning, colorimetry

## Abstract

Poly(3,4-ethylenedi-oxythiophene) (PEDOT) derivatives conducting polymers are known for their great electrochromic (EC) properties offering a reversible blue switch under an applied voltage. Characterizations of symmetrical EC devices, built on combinations of PEDOT thin films, deposited with a bar coater from commercial inks, and separated by a lithium-based ionic membrane, show highest performance for 800 nm thickness. Tuning of the color is further achieved by mixing the PEDOT film with oxides. Taking, in particular, the example of optically inactive iron oxide Fe_2_O_3_, a dark blue to reddish switch, of which intensity depends on the oxide content, is reported. Careful evaluation of the chromaticity parameters *L**, *a**, and *b**, with oxidizing/reducing potentials, evidences a possible monitoring of the bluish tint.

## 1. Introduction

In recent years, electrochromic devices (ECDs), known for their ability to modify their optical properties under an applied voltage, have received growing interest [1,2,3]. ECDs are based on a combination of materials assembled in various configurations either in a transmittive or reflective mode and find applications in smart windows, rearview mirrors, and displays [4,5,6]. The list of electrochromic materials include inorganic systems (e.g., WO_3,_ V_2_O_5,_ NiO [7,8,9] and Prussian blue [10]) and organic ones (e.g., viologens, polyaniline, and poly(3,4-ethylenedioxythiophene)). Compared with inorganic electrochromic materials, organic electrochromic materials have advantages such as a broader color palette, higher coloration efficiency, and greater optical contrast [11,12,13,14,15,16]. In addition, recent studies have demonstrated the suitability of a large number of techniques, including spray coating, ink jet, and doctor blading, for deposition of thin films of conducting polymers with good control of the process [17,18,19,20,21,22,23]. Indeed, the choice of the processing method will affect not only the charge to activate the device but also their stability. Among π-conjugated polymers, polypyrrole (PPy), polyaniline (PANI), and polythiophene (PTP) are the most widely studied. In practical use, most of the inks commercially available are PEDOT-based. Thus, aiming at transferring our findings from lab scale to production, we choose to focus our attention on PEDOT-based devices. More precisely, we recently demonstrated activation of PEDOT-based ECDs using a smartphone for detection as counterfeit labels. PEDOT-based devices offer a color change from colorless or light blue, depending on film thickness, to dark blue. One means to widen the range of available colors is to combine PEDOT with oxides in such a ratio that the oxide color becomes predominant and the role of PEDOT is more as a conductive agent. This approach is more dependent on the oxide contribution and may largely modify the process. Aiming at an easy way to enlarge the color range of the commercially available PEDOT-based ink, the addition of a small quantity of oxides is currently investigated in our group. Herein, we particularly focus on Fe_2_O_3_ addition offering a reddish modulation of the usual blue color of PEDOT. Prior to the oxide addition, a detailed investigation of the influence of the PEDOT thickness on the EC behavior is carried out utilizing lithium-based symmetrical devices.

## 2. Materials and Methods

### 2.1. Electrochromic Ink Formulation

The electrochromic inks were formulated from commercial PEDOT:PSS ink Agfa Orgacon EL-P5015 (called hereinafter PEDOT:PSS paste). This high-viscosity commercial paste (>100,000 mPa·s) made for screen printing was first homogenized with a three-roll mill and then diluted with ethanol to lower the viscosity as follows.

For the PEDOT:PSS ink, a mixing ratio of 40 wt % of PEDOT:PSS paste and 60 wt % of ethanol was prepared. The resulting dilute solution was stirred for 15 min at room temperature, then dispersed using an ultrasonic bath for 15 min and stirred again for 15 min.

For the PEDOT:PSS+Fe_2_O_3_ ink, commercial Fe_2_O_3_-based pigment was employed (Rouge 110 purchased from Ocres de France, Fe_2_O_3_ content >97.1%). The Fe_2_O_3_ weight percentages were calculated from the PEDOT:PSS paste mass. Four premixes were prepared: weight ratio of (PEDOT:PSS paste)/(Fe_2_O_3_) = (100−x)/x, with x = 2%, 2.5%, 3%, and 3.5%. These preparations were then diluted with ethanol with a mixing ratio of 40 wt % of (PEDOT:PSS + Fe_2_O_3_) paste and 60 wt % of ethanol. The resulting dilute was stirred for 15 min at room temperature, then dispersed using an ultrasonic bath for 15 min and stirred again for 15 min.

### 2.2. Thin Film Deposition and Display Processing

The films were deposited with a bar coater (K control from RK PrintCoat Instruments, Erichsen RK controle, Valence, France) onto ITO-coated (In_2_O_3_:Sn) glass substrates (commercialized by SOLEMS with a resistance of 30 Ω □^−1^) and then dried at 120 °C/5 min on a hot plate. In order to vary the thickness of the PEDOT:PSS films, several bars were used for the process, numbered from 1 to 7, corresponding to wet film coating thickness of, respectively, 6 µm, 12 µm, 24 µm, 40 µm, 50 µm, 60 µm, and 80 µm. For the PEDOT:PSS + Fe_2_O_3_ films, bar n°5 was used.

The displays were made in a symmetric configuration, that is, for each device the same PEDOT:PSS/ITO/glass stack was employed for both working and counter electrochromic electrodes. An electrolytic membrane was then used to glue these two electrodes and to ensure the ionic conduction. This membrane was prepared and coated as follows. A commercial lithium-ionic-liquid-based solution of lithium bis(trifluoromethanesulfonyl)imide in 1-ethyl-3-methylimidazolium bis(trifluoromethanesulfonyl)imide containing 40 wt % of PMMA dissolved in butanone (EmimTFSI:LiTFSI (9:1 Molar Ratio), 40 wt % PMMA in butanone, 99.9%, Solvionic, Toulouse, France) was mixed with TiO_2_ powder (Titanium (IV) Oxide ≥99% Sigma Aldrich, Merck KGaA, Darmstadt, Germany) as a white opaque pigment in weight ratio of 90/10. The resulting mixture was stirred for 15 min at room temperature, then dispersed using an ultrasonic bath for 15 min and stirred again for 15 min. The electrolyte mixture was then deposited by bar coating (bar n°8, wet thickness 100 µm) on both PEDOT:PSS/ITO/glass electrodes. After 30 s of butanone evaporation, the two pieces were assembled and pressed manually. The obtained average thickness for the electrolytic membrane was 70 µm ± 10 µm. The final stack for each display presented in this article was glass/ITO/PEDOT:PSS/(EmimTFSI:LiTFSI + PMMA + TiO_2_)/PEDOT:PSS/ITO/glass. Combining various thicknesses, a set of seven electrochromic displays was prepared for the thickness optimization and four displays for the Fe_2_O_3_ addition. 

### 2.3. Structure, Morphology, and Thickness Measurements

Powder X-ray diffraction (XRD) patterns were collected on a PANalitycal X’pert PRO MPD diffractometer (Malvern Panalytical, Almelo, The Netherlands) in Bragg–Brentano *θ*-*θ* geometry equipped with a secondary monochromator and X’Celerator multistrip detector. Each measurement was made within an angular range of *2θ* = 8–80°. The Cu-Kα radiation was generated at 45 kV and 40 mA (*λ* = 0.15418 nm). Thickness measurements were performed using a Veeco Dektak 6M Stylus Profilometer (Veeco Instruments Inc., New York, NY, USA). The average thicknesses obtained with bars numbered from 1 to 7 were respectively 109 nm, 202 nm, 354 nm, 550 nm, 730 nm, 880 nm, 1106 nm, with a measurement uncertainty of 10%. The morphology of the layers was investigated using a JEOL JSM-840 (operating at 15 kV) scanning electron microscope (JEOL JSM-840, JEOL SAV-Europe, Croissy sur Seine, France).

### 2.4. Electrochromic Measurements

Concerning the PEDOT:PSS single layer on ITO/glass study, the electrochemical analyses were carried out in a three-electrode cell configuration using a BioLogic SP50 potentiostat/galvanostat apparatus (BioLogic SP50, Seyssinet Pariset, France). The counter electrode and reference electrode consisted of a platinum foil and saturated calomel electrode, SCE (E_SCE_ = 0.234 V/NHE), respectively. The operating voltage was controlled between −1.3 V and +1.3 V for chronoamperometry analysis, in a commercial lithium-based ionic liquid, namely EmimTFSI:LiTFSI (9:1 Molar Ratio, Solvionic, 99.9%). All the electrochemical measurements were performed at room temperature. The optical transmittance of PEDOT:PSS thin films was measured in situ using a Varian Cary 5000 UV-vis-NIR spectrophotometer (Agilent, les Ulis, France) between 250 and 800 nm.

Concerning the PEDOT:PSS displays, electrochemical analysis was performed in a two-electrode configuration, using a BioLogic SP50 potentiostat/galvanostat apparatus (BioLogic SP50, Seyssinet Pariset, France). Colorimetry analysis was carried out using a Konica Minolta CM-700D spectrophotometer (Konica Minolta Sensing Europe B.V., Roissy, France), allowing the direct determination of colorimetric parameters of the CIE (*L*a*b**) color space.

## 3. Results and Discussion

### 3.1. Optimization of the PEDOT:PSS Thickness

Figure 1 shows the electrochemical performances of the displays obtained from PEDOT:PSS thin films of different thicknesses. In Figure 1a, the second cycles of cyclic voltammetry measurements (*CV*) are presented. The operating voltage was controlled between −1.6 V and +1.6 V at a scan rate of 40 mV·s^−1^. A symmetric behavior is observed for both reduction and oxidation, due to the symmetric geometry of the stack. The rectangular-like shape of the *CV* is typical of a pseudo capacitive behavior. As expected, the capacity increases with the increase of the PEDOT:PSS thickness. The switching kinetic of the displays was investigated by chronoamperometry measurements. Figure 1b shows the current response (*j*-*t*) of the displays for potential steps of +1.6 V/30 s and −1.6 V/30 s. A symmetric behavior is also observed between oxidation and reduction. The display current response increases with thickness from 0.4 s for the thinnest film, to less than 1 s for the thickest one. Complementary information about the switching time calculation is given in Appendix A, in which the switching time is deduced from 95% drop of the current density. The increase of the film thickness leads to an increase of the capacity *Q* of the electrode, therefore of charge density, and causes an expected increase of the current response time. In addition, the nonlinearity of the switching time versus thickness well agrees with a decrease in the PEDOT resistance when increasing the thickness [24].

Figure 2 shows the photographs of the displays taken during the chronoamperometry measurements presented above. A clear color gradient is observed with the increase of thickness, in oxidized (bleached) state and mainly in reduced neutral (colored) state. Oxidized state graduates from whitish blue to light blue whereas the reduced state color graduates from light blue to dark blue. In conclusion, the thickness of the electrochromic layer is an easy way to tune the color of the display, as shown on the corresponding measured *L*a*b** parameters mentioned next to each image.

In order to evaluate the impact of the PEDOT:PSS thickness on the color contrast and determine the optimum thickness, colorimetric measurements of *L**, *a**, *b** chromaticity parameters were carried out during *CA* measurements for 30 cycles (Figure 3a). The optical contrast, herein the color contrast, Δ*E** was then calculated from the *L**, *a**, *b** values from the following formula: (1)$$\Delta E^{*}={\left[{\left(L_{red}^{*}-L_{ox}^{*}\right)}^{2}+{\left(a_{red}^{*}-a_{ox}^{*}\right)}^{2}+{\left(b_{red}^{*}-b_{ox}^{*}\right)}^{2}\right]}^{1/2}$$

Here, $L_{red}^{*}$, $a_{red}^{*}$, and $b_{red}^{*}$ represent color space parameters at the reduced state, and $L_{ox}^{*}$, $a_{ox}^{*}$, and $b_{ox}^{*}$ correspondingly at the oxidized state. For films with a thickness below or equal to 200 nm, the color contrast continuously decreases, whereas for higher thickness, the contrast first increases for the 10 first cycles and then slightly decreases for the next 20 cycles (Figure 3a). In good agreement with the visual aspect of Figure 2, showing a more pronounced bluish color from the 354 nm thickness towards thicker film for the reduced state, a large increase in the color contrast is visible from this thickness threshold. 

Figure 3 presents Δ*E** as a function of the film thickness for the 10th *CA* cycle. Overall, the color contrast increases with increasing thickness from 109 to 1106 nm, with a maximum of Δ*E** value around 36 at 880 nm. The “bell” shape of this plot is in good agreement with previous studies in the literature [17,25]. Compared to Kawahara’s study, also on PEDOT:PSS electrochromic displays, the optimal thickness is found at higher thickness in our case (i.e., 500–600 nm vs. 700–800 nm) [25]. However, Kawahara and coworkers always used the same thickness for the counter electrode in each of their devices that could change the color perception. These authors also emphasized the importance of having a transparent enough oxidized state favoring thinner thickness.

So as to further investigate the properties of the optimum thickness of PEDOT:PSS (i.e., 880 nm), display, electrochemical, and optical properties of the single layer deposited on ITO/glass were studied. On Figure 4a, chronoamperometry measurements, performed in a three-electrode configuration using ionic-liquid-based (0.3 M) Li-TFSI in EMI-TFSI as a supporting electrolyte, Platinum as counter electrode and Saturated Calomel Electrode as reference electrode (SCE) are presented. Potential was applied for 20 s at −1.3 V for reduction and 20 s at +1.3 V for oxidation. Both response times were approximately 3 s. The same potentials were applied for several minutes for in situ optical transmittance measurements. The corresponding spectra for the bleached and the colored states are shown in Figure 4b. The associated transmittance values at 600 nm are, for the bleached state, *T_b_* ≈ 44% and, for the colored state, *T_c_* ≈ 7%, corresponding to a total optical transmittance modulation of ∆*T* ≈ 37%. The electrochromic performance is commonly characterized by the optical density (*OD*) represented by the logarithmic ratio of the transmittance in the bleached state (*T_b_*) to the transmittance in the colored state (*T_c_*) and the coloration efficiency (*CE*) defined as the change in optical density per unit inserted charge. The Δ*OD* and *CE* are calculated using the following two relations:ΔOD = log(Tb/Tc)(2)
*CE* = Δ*OD*/*Q*(3)

Here, *Q* is the amount of charge transferred per unit area. The calculated optical density Δ*OD* = 0.76 and the amount of charge *Q* = 0.01451 C.cm^−2^ leads to a coloration efficiency value of 52.4 cm^2^·C^−1^ The rather low *CE* value as compared to the ones reported in the literature [1,14,15,17] may originate from the wide potential range leading to large current values measured using chronoamperometry while the capacity is usually deduced from dynamic measurements using cyclic voltammetry.

In order to study the long-term performances of the display with the optimum PEDOT:PSS thickness, color contrast was measured during chronoamperometry cycling. An oxidizing potential of +1.6 V was applied for 30 s, and reducing potential of −1.6 V for 30 s, during 500 cycles. In situ colorimetric measurements were carried out every 50 cycles. The evolution of the color contrast upon cycling is presented in Figure 5. It should be noted that this display was already cycled around 50 times before this measurement, explaining the lower starting value of Δ*E** of 27 as compared to the one of 34 (Figure 3b). During the 100 first cycles, the optical contrast is stable and then slowly decreases to a stable value of Δ*E** ≈ 20 at 500 cycles. 

### 3.2. Tuning of the PEDOT:PSS Color by Fe_2_O_3_ Pigment Addition

In the earlier section, we demonstrated that the electrochromic layer thickness could enable the tuning of the PEDOT:PSS tint. Another original way to change and adapt the PEDOT:PSS color is to add an inert color pigment to the electrochromic layer. Herein we present the results obtained by adding Fe_2_O_3_ red pigment into the PEDOT:PSS ink from 2 wt % to 3.5 wt %. The pigment ratios were chosen after previous empirical trials. As explained in Section 2.1, these percentages correspond to the mixing ratio of the pigment with the PEDOT:PSS paste. Therefore, after the thermal drying of the layer, the Fe_2_O_3_ ratio increases significantly. For convenience, the devices will be designated by the mixing ratio thereafter. 

It is well known that Fe_2_O_3_ exhibits many different crystalline forms. The XRD pattern of the Fe_2_O_3_-based pigment corresponds to the *α*-Fe_2_O_3_ phase (Hematite) (Figure 6). In the literature, Fe_2_O_3_ electrochromic properties have been the subject of debates [26,27,28,29,30,31,32,33]. Gutiérrez and Beden have confirmed that FeOOH is responsible for the electrochromic effect, and also identified weak absorption peak under potential that could be attributed to *γ*-Fe_2_O_3_ and *α*-Fe_2_O_3_ [27]. However, later studies found that *α*-Fe_2_O_3_ did not present any electrochromic properties [29,31]. More recently, Garcia-Lobato proposed new insight into the electrochromic properties, showing that *α*-Fe_2_O_3_ can be, in some aqueous electrolytic conditions, a starting phase for the transformation to a Fe(OH)_2_ phase and subsequently *γ*-Fe_2_O_3_, topotactic redox reactions responsible for color changes [32,33]. Therefore, in the present study, considering the literature background, the hematite Fe_2_O_3_-based pigment can be considered electrochromically neutral.

The average thickness measured by mechanical profilometer for Fe_2_O_3_ contents of 2%, 2.5%, 3%, and 3.5% is 814 nm, 918 nm, 1090 nm, and 1235 nm, respectively. As discussed above, the effective Fe_2_O_3_ contents in the films are much higher, explaining the large increase of thickness for films deposited initially using a similar bar (i.e., bar n°5, wet thickness 50 µm).

Figure 7 shows the SEM micrographs of the Fe_2_O_3_/PEDOT:PSS composite film and the one of the PEDOT:PSS single layer The Fe_2_O_3_ particles, of few hundreds of nanometers, are well dispersed in the PEDOT:PSS matrix. A significant increase of the film roughness from Ra = 74 nm (PEDOT) to 200 nm (PEDOT +2.5% Fe_2_O_3_) was measured by profilometry. 

Symmetrical displays were built from combining PEDOT:PSS/% Fe_2_O_3_ thick films. The cyclic voltammograms of the displays show an evolution of the shape with the Fe_2_O_3_ pigment addition (Figure 8a). Indeed, an extra electrochemical response is measured in oxidation between −0.5 V and 0.8 V and symmetrically during reduction in the same potential range. The electrochemical response is not associated with an electrochromic phenomenon, as shown in Figure 8b, as a very low color contrast was measured between 0.6 V and −0.6 V. Campet et al. reported that Fe_2_O_3_ thin film electrodes exhibit an electrochemical response attributed to Li^+^ electrochemical (de)insertion in very fine-grained film or when the microcrystallites were finely dispersed in conductive polymers [28], that could explain this electrochemical response with the addition of Fe_2_O_3_. PEDOT:PSS oxidation and reduction peaks remain at the same potentials, respectively at +1.6 V and −1.6 V. Therefore, color switching of the displays was studied around these potentials in chronoamperometry. 

An example of the obtained colors at the oxidized and reduced states of PEDOT:PSS with 2% Fe_2_O_3_ content is shown in Figure 9a. The photographs were taken during CA measurements: on top images in three-electrode configuration at *V* = +1.3 V and *V* = −1.3 V using ionic-liquid-based (0.3 M) Li-TFSI in EMI-TFSI as a supporting electrolyte, Platinum as counter electrode, and Saturated Calomel Electrode as reference electrode, on bottom images in display configuration at *V* = +1.6 V and *V* = −1.6 V. On the oxidized state, due to the high tinting strength of Fe_2_O_3_ on the light blue color of PEDOT:PSS, the Fe_2_O_3_ addition induces a red color in both configurations. On the reduced state, the film color switches to dark blue; in this case, the PEDOT:PSS-reduced color takes over the Fe_2_O_3_ pigment. 

The tint of the red and blue colors can then be tuned with the reducing potential and the Fe_2_O_3_ content, as shown in Figure 9b–d on the *L*a*b** color space parameters. The colorimetric measurements were performed during CA measurements on displays with various Fe_2_O_3_ rates. Each time, the oxidation potential was applied at +1.6 V/30 s and the reduction potential was alternatively applied at +1.4 V/30 s to +1.8 V/30 s increasing by 0.1 V steps.

For each Fe_2_O_3_ content, the color contrast is similar and it increases with the reduction voltage, correlating the decrease of the *L** parameter with the reduction potential. The color tints were studied by representation in the *b** versus *a** plan (Figure 9d). In one hand, by increasing the Fe_2_O_3_ contents, the oxidized state can be adjusted to deeper red colors. The chroma of the color *C** was calculated as follows: (4)$$C^{*}=\sqrt{\left(a^{*2}+b^{*2}\right)}$$

The increase of Fe_2_O_3_ from 2% to 3.5% increases the *C** value from 17 to 27, with a haze of 45°. On the other hand, the reduced state can be adjusted to darker blue colors by decreasing the Fe_2_O_3_ contents, increasing the chroma value from 6 to 14 at −1.8 V, along the *−b** axis. Moreover, by varying the reduction potential, the tint of the dark blue can be finely tuned to adapt the desired color tint, as shown for the 3% in the inset of Figure 9d.

## 4. Conclusions

PEDOT-PSS films were successfully deposited using a bar coater with thickness ranging from 100 to 1100 nm. Careful investigation of the relationship between the PEDOT:PSS layer thickness and electrochromic performances of symmetrical devices, built by separating the PEDOT:PSS layer with lithium electrolyte membrane, identified an optimized thickness of approximatively 880 nm. Meanwhile, the thickness modulation led to a nice control of the bluish tint in the reduced state. Further achievement in tuning the color of the EC device was suggested by oxide addition. EC device based on the mixture of PEDOT:PSS and Fe_2_O_3_ opens up the color modulation from red to blue. In the field of applications, our approach is a promising route for tuning the color of products based on PEDOT:PSS-based ink commercially available in large scale. 

## Figures and Tables

**Figure 1 polymers-11-00179-f001:**
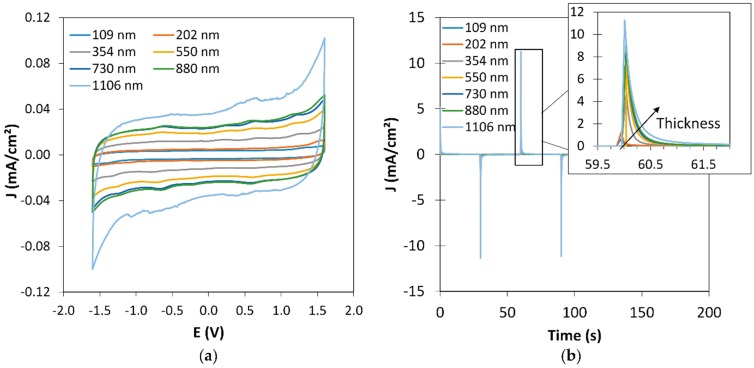
(**a**) Cyclic voltammograms of the electrochromic symmetrical displays built from PEDOT:PSS thin films with varying thickness from 109 nm to 1106 nm, (**b**) chronoamperograms (*CA*) of the electrochromic symmetrical displays built from different PEDOT:PSS thicknesses. Potentials of +1.6 V and −1.6 V were applied for 30 s alternatively.

**Figure 2 polymers-11-00179-f002:**
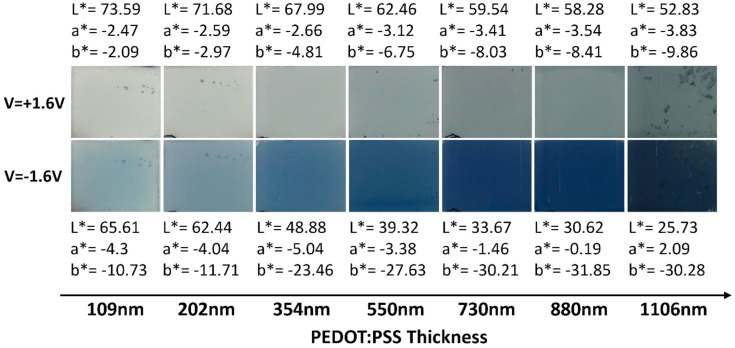
Photographs of the electrochromic symmetrical displays built from PEDOT:PSS films with different thicknesses, showing an oxidized state for a potential of +1.6 V and a reduced (colored) state for a potential of −1.6 V. The corresponding measured *L*a*b** parameters are mentioned at the top and bottom of the image for the oxidized (bleached) state and the neutral (colored) state, respectively.

**Figure 3 polymers-11-00179-f003:**
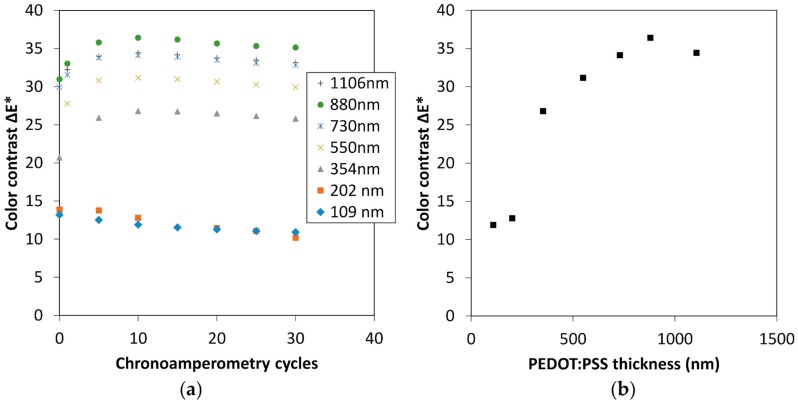
(**a**) Color contrast $\Delta E^{*}$ as a function of the chronoamperometry cycles of electrochromic displays built from PEDOT:PSS thin films with varying thickness from 109 nm to 1106 nm, (**b**) color contrast as a function of PEDOT:PSS thickness, after 10 cycles of chronoamperometry. For both graphs, applied potentials were +1.6 V for 20 s, and 1.6 V for 20 s.

**Figure 4 polymers-11-00179-f004:**
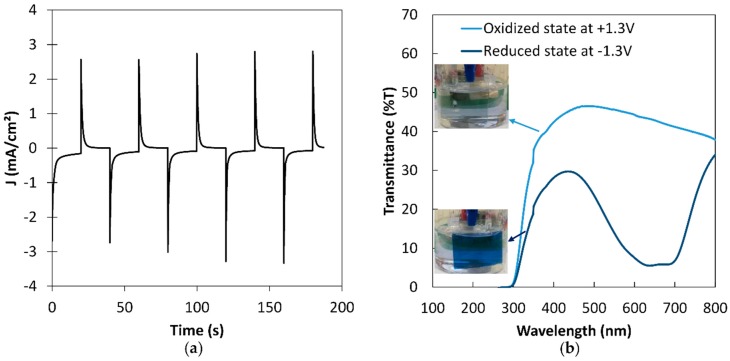
(**a**) Chronoamperograms in a three-electrode cell configuration at −1.3 V for 20 s and +1.3 V for 20 s of the PEDOT:PSS layer at the optimized thickness of 880 nm, (**b**) corresponding optical transmittance spectra at oxidized and reduced states of PEDOT:PSS layer at the optimized thickness of 880 nm (the pictures of the thin films during the test are presented in the inset).

**Figure 5 polymers-11-00179-f005:**
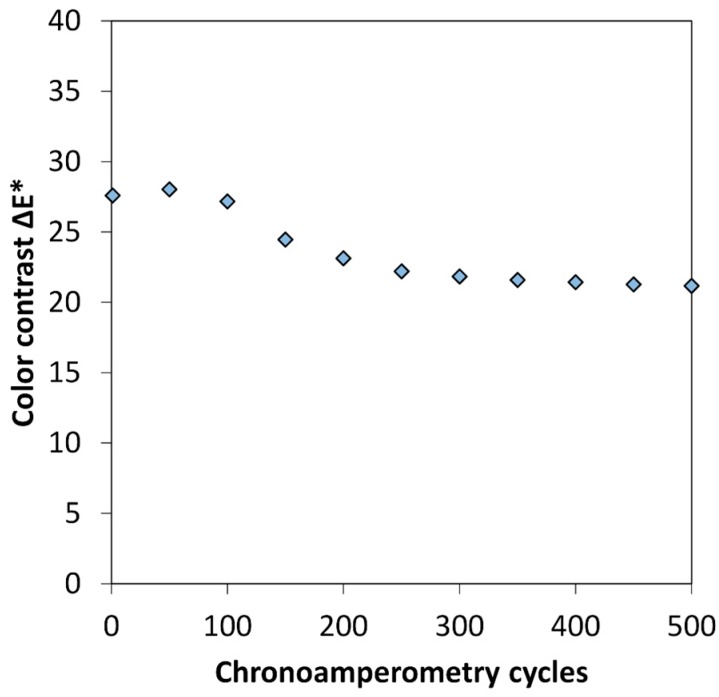
Color contrast Δ*E** as a function of the number of chronoamperometry cycles, measured on an electrochromic symmetrical display built from 880 nm PEDOT:PSS thick film.

**Figure 6 polymers-11-00179-f006:**
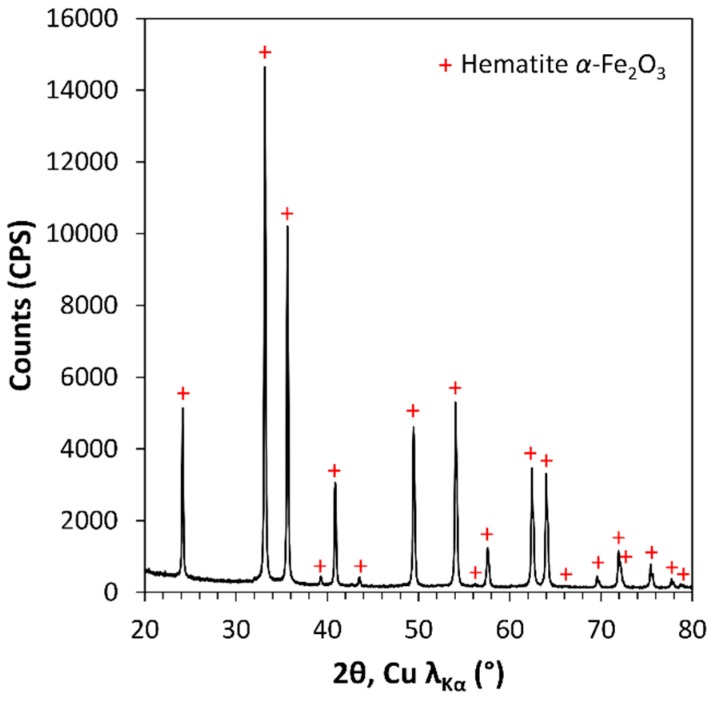
XRD pattern of the Fe_2_O_3_ pigment. The red crosses indicate α-Fe_2_O_3_ structure diffraction peaks (space group R-3C).

**Figure 7 polymers-11-00179-f007:**
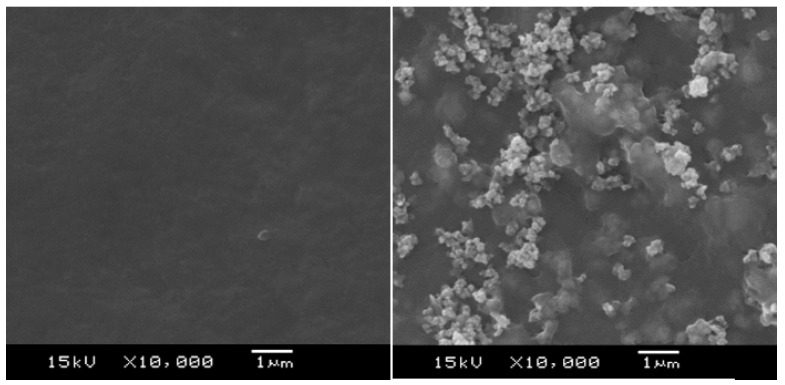
Top view SEM images of the surface morphology of two films. On the left, PEDOT:PSS film with an average thickness of 550 nm; on the right, PEDOT:PSS + 2.5% of Fe_2_O_3_ film with an average thickness of 814 nm. Both films were deposited using the same wirebar (i.e., bar n°5, wet thickness 50 µm).

**Figure 8 polymers-11-00179-f008:**
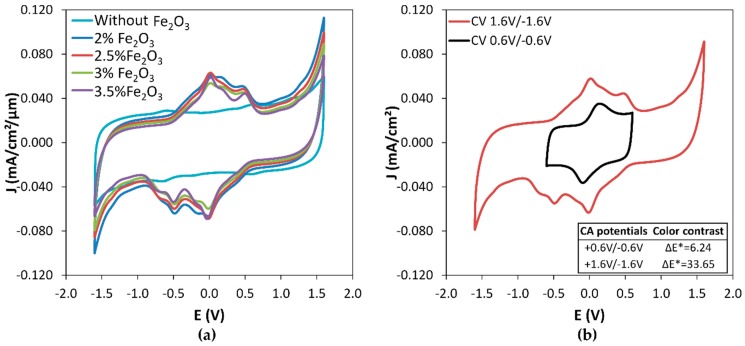
(**a**) Cyclic voltammograms (2^nd^ cycle) of the electrochromic displays obtained from PEDOT-PSS + x Fe_2_O_3_ thin film with varying x from 2 wt % to 3.5 wt % compared to PEDOT:PSS without Fe_2_O_3_, (**b**) cyclic voltammograms of the electrochromic displays with PEDOT:PSS +2.5% Fe_2_O_3_ at different potential ranges.

**Figure 9 polymers-11-00179-f009:**
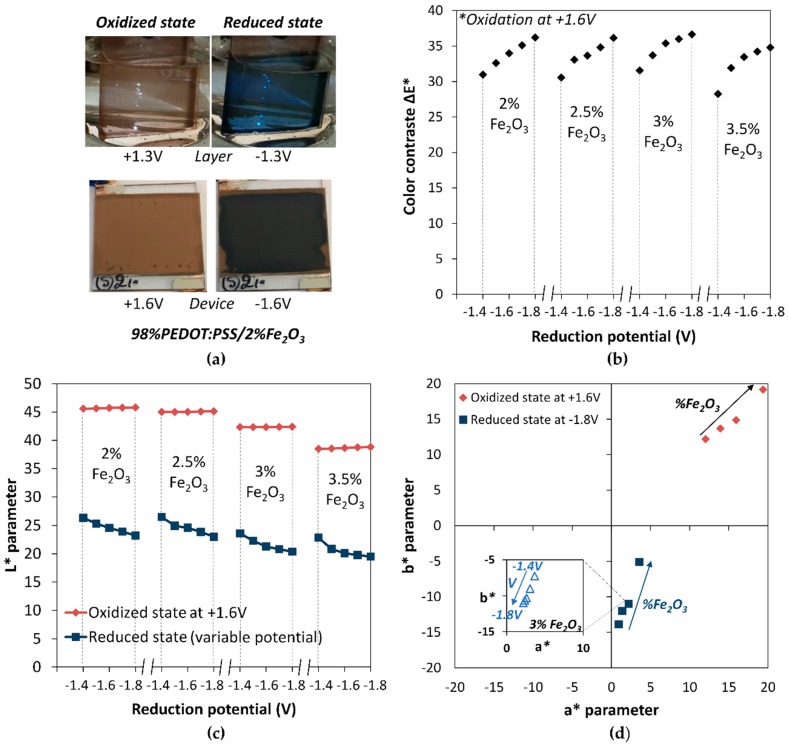
This figure shows the tuning of the *L*a*b** parameters by adjusting the Fe_2_O_3_ content or the reduction voltage. (**a**) Image of the oxidized and reduced state of a 98%PEDOT:PSS/2%Fe_2_O_3_ thin layer and display, changing from red to blue color. (**b**) *L** parameter as a function of the reduction voltage for different Fe_2_O_3_ contents. (**c**) Tuning of *a*b** parameters by increasing Fe_2_O_3_ content for oxidized state at 1.6 V and reduced state at −1.8 V (**d**); inset shows the tuning of the blue shade by varying the reduction voltage.

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
