# Peer review of "Color Tuning by Oxide Addition in PEDOT:PSS-Based Electrochromic Devices"

_polymers, 2019, doi:10.3390/polym11010179_

Round 1

Reviewer 1 Report

This is a good, industrially-relevant paper on electrochromic materials. The introductory material, analysis and the references are useful and appropriate. 

The only issues with this paper are associated with English phrasing / word choices. My suggestions for changes are below (there may be others identified by the editors).

Line 9: I suggest inserting the word "are" between the words "derivatives" and "conducting" and inserting the word "and" between the words "polymers" and "are"

Line 24: Insert the word "and" between the word "mirrors" and "displays"

Line 30: Replace the word "well" with another word like "good", "excellent" or "adequate" to describe the level of process control

Lines 33 to 35: The sentence beginning with "However in practical use..." is awkward and confusing...I think that you want to say that "PEDOT-based inks are available in commercial quantities and therefore would be a good choice for electrochromic applications." but I'm not completely sure.

Line 37: I think that what was meant was that "...PEDOT-based ECD using a smartphone to detect counterfeit labels." the words "thanks to" should be changed to the word "using" and the word "detecting" should be added (if that was the application).

Line 39: Change "wider" to "widen"

Lines 40 & 41: Change "Such approach reveals to be more..." to "This approach is more..." 

Line 51: Change "printed" to "printing"

Line 54: Insert the word "solution" between the words "dilute" and "was"

Line 58: Change "pre-mix" to "pre-mixes" and change the words "in that respect" to "with" 

Line 61: Insert the word "solution" between the words "dilute" and "was"

Line 96: Change the word "were" to "was"

Line 158: Change the words "used always" to "always used"

Line 159: Change the word "device" to "devices" and change "color perception" to "perceived color" and change the word "underline" to "emphasize"

Line 169: The symbols "Tb" and "Tc" are used without explanation...the explanation is in line 172. I suggest moving the explanatory phrases before the first use of "Tb" and "Tc"

Line 204: Insert the word "the" between the words "earlier" and "section"

Line 249: Delete the word "subsequently"

Lines 263 & 294: Choose a less qualitative word or phrase than the word "nice" such as the phrase "the ability to"

Line 278: Change the phrase "On another hand" to the more commonly used "On the other hand"

Line 290: Inert the word "a" between the words "using" and "bar" 

Line 293: Change the phrase "concludes on" to the word "identified"

Author Response

Reviewer #1:
This is a good, industrially-relevant paper on electrochromic materials. The introductory material, analysis and the references are useful and appropriate.
The only issues with this paper are associated with English phrasing / word choices. My suggestions for changes are below (there may be others identified by the editors).

Line 9: I suggest inserting the word "are" between the words "derivatives" and "conducting" and inserting the word "and" between the words "polymers" and "are"

Unfortunately, I disagree on this point and left it the way it is.

Line 24: Insert the word "and" between the word "mirrors" and "displays"

OK
Line 30: Replace the word "well" with another word like "good", "excellent" or "adequate" to describe the level of process control

OK
Lines 33 to 35: The sentence beginning with "However in practical use..." is awkward and confusing...I think that you want to say that "PEDOT-based inks are available in commercial quantities and therefore would be a good choice for electrochromic applications." but I'm not completely sure.

We thank the reviewer who did indeed understand the meaning of the sentence. In the revised version, the sentence was simplified and rephrased as follows : In practical use, most of the inks commercially available are PEDOT based. Thus, aiming at transferring our findings from lab

Line 37: I think that what was meant was that "...PEDOT-based ECD using a smartphone to detect counterfeit labels." the words "thanks to" should be changed to the word "using" and the word "detecting" should be added (if that was the application).

OK 

Line 39: Change "wider" to "widen"

OK
Lines 40 & 41: Change "Such approach reveals to be more..." to "This approach is more..."

. In practical use, most of the inks commercially available are PEDOT based. Thus, aiming at transferring our findings from lab

Line 51: Change "printed" to "printing"

OK
Line 54: Insert the word "solution" between the words "dilute" and "was"

OK
Line 58: Change "pre-mix" to "pre-mixes" and change the words "in that respect" to "with"

OK
Line 61: Insert the word "solution" between the words "dilute" and "was"

OK
Line 96: Change the word "were" to "was"

OK
Line 158: Change the words "used always" to "always used"

OK
Line 159: Change the word "device" to "devices" and change "color perception" to "perceived color" and change the word "underline" to "emphasize"

OK
Line 169: The symbols "Tb" and "Tc" are used without explanation...the explanation is in line 172. I suggest moving the explanatory phrases before the first use of "Tb" and "Tc"

OK
Line 204: Insert the word "the" between the words "earlier" and "section"

OK
Line 249: Delete the word "subsequently"

OK
Lines 263 & 294: Choose a less qualitative word or phrase than the word "nice" such as the phrase "the ability to"

OK
Line 278: Change the phrase "On another hand" to the more commonly used "On the other hand"

OK
Line 290: Inert the word "a" between the words "using" and "bar"

OK
Line 293: Change the phrase "concludes on" to the word "identified"
OK

Reviewer 2 Report

The manuscript entitled "Color tuning by oxide addition in PEDOT based electrochromic devices" can be considered for publication in Polymers. Please take into consideration the following comments prior publication.

1. The authors illustrate PEDOT films in the title, abstract and introduction sections. However, they depict the properties of PEDOT:PSS films in the materials and methods, results and discussion, and conclusion sections. The name of polymer film should be consistent.  

2. L99, define ENH of “ESCE = 0.234 V / ENH”.

3. L74, “1-Ethyl-3-methylimidazolium” -> “1-ethyl-3-methylimidazolium”.

4. L145, the reference of equation (E*) should be given.

5. The numbers of equations in L145, L175-176 and L276 should be assigned.

6. L175, “b” and “c” of “ΔOD = log(Tb/Tc)” should be subscripts.

7. L178, the coloration efficiency value should be 52.4 cm2 C−1.

8. The journal, year, volume and page should be assigned in reference 28.

9. The wavelengths of electrochromic measurements for Figs. 3 and 5 should be given.

10. L231-232, the authors illustrate “Fe2O3 particles, of few hundreds of nanometers are well dispersed in the PEDOT:PSS matrix”. Could you offer other data to prove this matter?

Author Response

Reviewer #2:
The manuscript entitled "Color tuning by oxide addition in PEDOT based electrochromic devices" can be considered for publication in Polymers. Please take into consideration the following comments prior publication.
1. The authors illustrate PEDOT films in the title, abstract and introduction sections. However, they depict the properties of PEDOT:PSS films in the materials and methods, results and discussion, and conclusion sections. The name of polymer film should be consistent.  

We thank the reviewer for his comments. The title has been modified including PEDOT-PSS.

2. L99, define ENH of “ESCE = 0.234 V / ENH”.

We thank the reviewer for pointing out this mistake. ENH was replaced by NHE for Normal Hydrogen Electrode.

The standard hydrogen electrode (abbreviated SHE), also called normal hydrogen electrode (NHE), is a redox electrode which forms the basis of the thermodynamic scale of oxidation-reduction potentials.

3. L74, “1-Ethyl-3-methylimidazolium” -> “1-ethyl-3-methylimidazolium”.

OK
4. L145, the reference of equation (∆E*) should be given.

No specific reference exists. The (∆E*) definition can be also found in  Ref 7 and 8.

 5. The numbers of equations in L145, L175-176 and L276 should be assigned.

OK
6. L175, “b” and “c” of “ΔOD = log(Tb/Tc)” should be subscripts.

OK
7. L178, the coloration efficiency value should be 52.4 cm2 C−1.

OK

8. The journal, year, volume and page should be assigned in reference 28.

OK
9. The wavelengths of electrochromic measurements for Figs. 3 and 5 should be given.

We did not include the wavelengths for Fig. 3 and 5 as the ∆E* is deduced on the full spectrum (From 250 nm to 800 nm).  Those values were added in the experimental section.

10. L231-232, the authors illustrate “Fe2O3 particles, of few hundreds of nanometers are well dispersed in the PEDOT:PSS matrix”. Could you offer other data to prove this matter

The proof of the dispersion of Fe2O3 is mainly deduced from  SEM analysis (Fig. 7 of the revised version). 
